# A Novel Multi-Scale Transformer for Object Detection in Aerial Scenes

**Guanlin Lu, Xiaohui He *, Qiang Wang, Faming Shao, Hongwei Wang and Jinkang Wang**

Department of Mechanical Engineering, College of Field Engineering, Army Engineering University, PLA, Nanjing 210007, China; luguanlin@aeu.edu.cn (G.L.); wangqiang@aeu.edu.cn (Q.W.); shaofaming@aeu.edu.cn.com (F.S.); wanghongwei@aeu.edu.cn (H.W.); wangjingkang@aeu.edu.cn (J.W.)
* Correspondence: gcbhxh@aeu.edu.cn

**Abstract:** Deep learning has promoted the research of object detection in aerial scenes. However, most of the existing networks are limited by the large-scale variation of objects and the confusion of category features. To overcome these limitations, this paper proposes a novel aerial object detection framework called DFCformer. DFCformer is mainly composed of three parts: the backbone network DMViT, which introduces deformation patch embedding and multi-scale adaptive self-attention to capture sufficient features of the objects; FRGC guides feature interaction layer by layer to break the barriers between feature layers and improve the information discrimination and processing ability of multi-scale critical features; CAIM adopts an attention mechanism to fuse multi-scale features to perform hierarchical reasoning on the relationship between different levels and fully utilize the complementary information in multi-scale features. Extensive experiments have been conducted on the FAIR1M dataset, and DFCformer shows its advantages by achieving the highest scores with stronger scene adaptability.

**Keywords:** object detection; aerial object detection; deep learning; multi-scale object detection; vision transformer



## 1. Introduction

In recent years, aerial images have become an essential data source in earth remote sensing because they can provide a large amount of information and are easy to access and real-time solid [1]. They can meet the requirements of practical tasks such as resource exploration [2], environmental monitoring [3], urban planning [4], and precision agriculture [5]. With the application of computer vision detection in remote sensing, aerial object detection has become a fundamental and active research topic. However, considering the characteristics of aerial images, effectively detecting objects in aerial scenes is still challenging.

The significant difference in the inherent scale of object instances mainly leads to the contrast of object scale in the image scene. Unlike the ground scene, the aerial scene has a long sight distance and a large field of vision, and the scale differences of many types of instances in the background are significant. In addition, especially for the same target, it is related to the change in the image acquisition distance [6]. The significant shift in image acquisition distance in a large-scale aerial scene will lead to a certain extent object scale variations. The farther the distance, the smaller the object. As shown in Figure 1.

Besides, aerial images are usually collected from a bird's eye view. The interferences in the large-field scene and the complex spatial distribution of ground objects confuse the object features in the background. In this case, different objects may show similar semantic features, while the semantic features of the same category of objects may differ significantly, as illustrated in Figure 2.

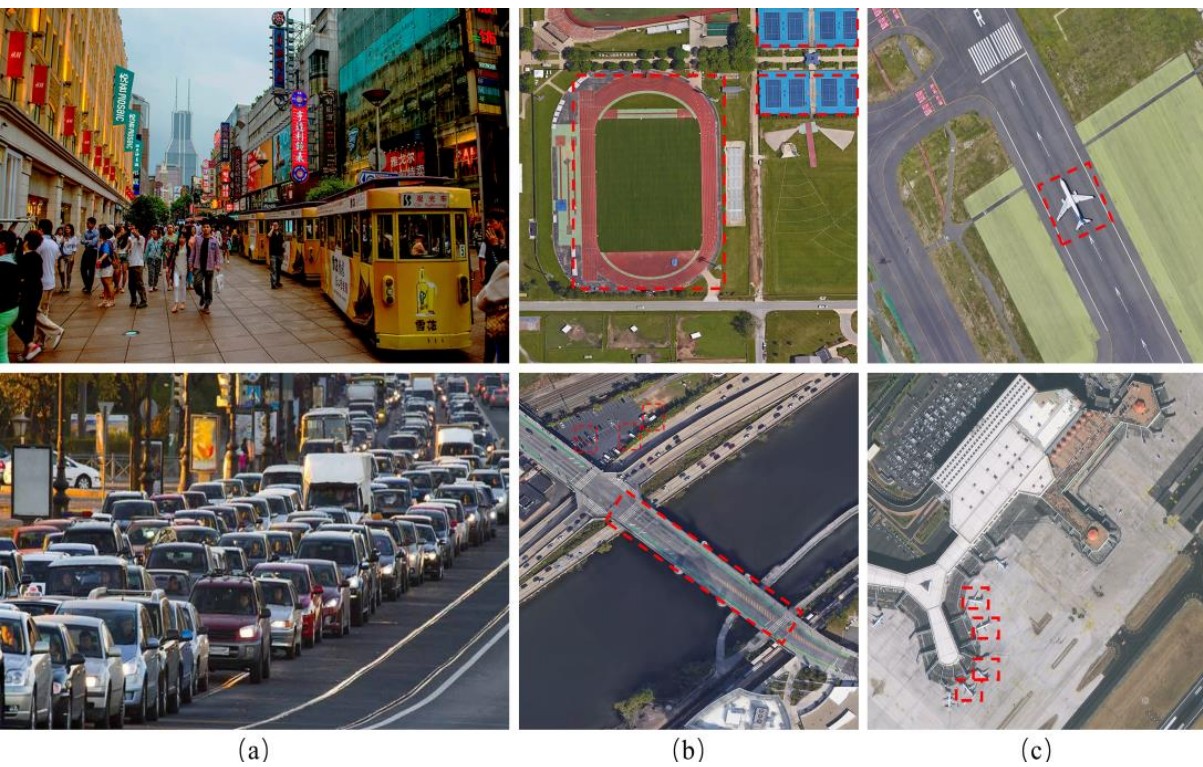

**Figure 1.** Comparing the scale variation in nature scenes: (**a**) with those in aerial scenes (**b**,**c**). The scales of different instances in group; (**b**) differ significantly, while the same instance in group; (**c**) varies considerably.

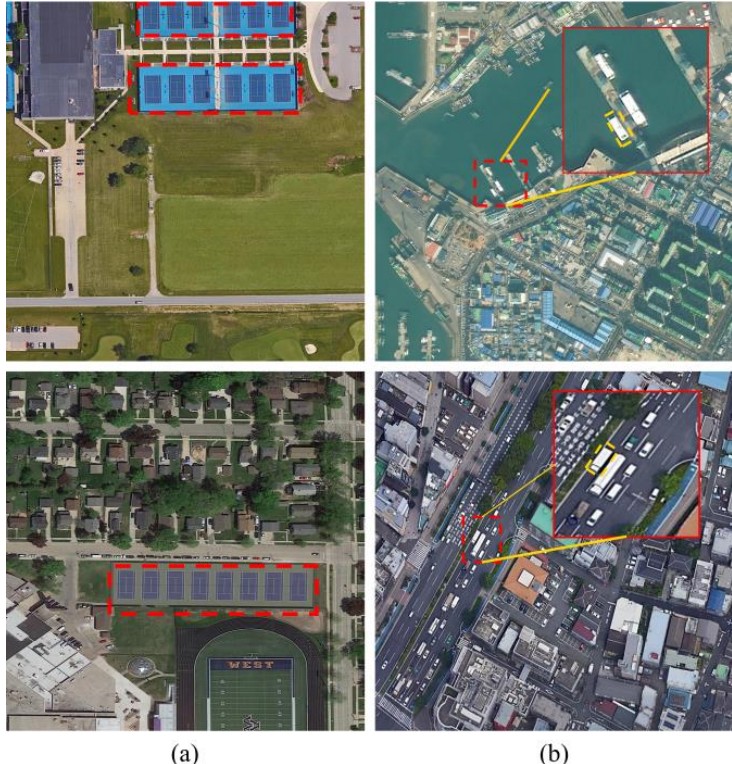

**Figure 2.** The challenges of object detection in aerial scenes. The red dashed box in group: (**a**) shows that the same objects category is significantly different in different scenes. In contrast, the yellow dashed box in group; (**b**) shows that different object categories are similar in different scenes.

The core of aerial object detection is to obtain the distinguishing features of objects in different scenes. Generally, there are three methods to obtain the features. In the early research, handcraft features are the core element of aerial object detection. Researchers directly use simple attributes of objects (color [7], contour [8], texture [9]) and some carefully designed feature descriptors (scale-invariant feature transformation and oriented gradient histogram) to distinguish objects. These handcraft features often perform well in simple scenes, but they cannot well depict objects in complex scenes. Meanwhile, researchers adopt feature coding methods [10–12] to encode color, contour, or texture features to deeply capture distinguishing features and mine semantic information, but the performance is limited and cannot meet the needs of detection in complex scenes.

In recent years, data-driven deep learning algorithms have made remarkable achievements in object detection [13–15], and a series of CNN-based aerial detection methods have been proposed. To overcome the challenges (i.e., large-scale variations and confused distinguishing features) in aerial object detection, Yang et al. [16] proposed a detection framework (ClusDet) for regional object clustering. It can significantly improve the detection performance of tiny objects, implicitly model a priori context information, and dramatically reduce the inference time. Further, to overcome the challenge of tiny scale and low signal-to-noise ratio of aerial objects, Wang et al. [17] presented a dataset (AI-TOD) for detecting tiny objects in aerial scenes. Based on this, a multiple center points learning-based detection network (M-CenterNet) was proposed. Different from the compound network structure of ClusDet, Li et al. [18] designed a detection framework (DMNet) based on the region clipping of the density map, which improved the detection accuracy. Meanwhile, Deng et al. [19] designed an end-to-end global-local adaptive network (GLSAN) to solve the problem of dense small objects and uneven distribution. The global-local fusion strategy was integrated into a progressive scale-change framework, and the global contour information and local detail information were utilized to enhance the robustness of the network against the scale variations of objects. Unlike the previous methods that mainly focus on improving the accuracy but ignore the memory and computation cost, Li et al. [20] proposed a detection framework called CorrNet. CorrNet adopts the lightweight VGG16 as the backbone network and adopts the coarse-to-fine strategy to promote the light development of the aerial object detection algorithm.

Although the CNN-based networks have made significant progress, the localization of convolution operation limits its ability to obtain the global information of large-scale and complex aerial scene images. The pooling operation with the receptive field expansion dilutes the feature details. Motivated by the excellent performance of the transformer [21] in natural language processing, researchers began to apply the transformer to visual recognition. Dosovitskiy et al. [22] transformed the image into patch sequence processing, designed the vision transformer (ViT), and achieved good performance in benchmark classification recognition. Compared with convolution in object detection, the transformer can emphasize the long-term dependencies between image patches and reserve abundant spatial information through the self-attention of multi-heads [23]. Therefore, researchers attempt to employ transformers to achieve accurate detection in complex aerial scenes. Li et al. [24] combined CNN with a transformer to design a network with an encoding decoding structure called TRD, which achieves good performances on NWPU VHR-10 [25] and DIOR [26] datasets. By combining convolution and transformer, Zhang et al. [27] designed a multi-scale network termed ViT-YOLO. It shows more substantial semantic resolution, effectively alleviates category confusion, and significantly enhances the detection performance of aerial objects. Similarly, Zhu et al. [28] proposed a detector called TPH-YOLOv5 based on transformer structure, which achieves good scene capture performance and impressive interpretability.

To sum up, attributed to the advantages of transformers in obtaining more context information and learning diversified feature representation, embedding transformers into the detection framework has the potential to overcome the challenges in aerial object detection.

This study proposes an effectively multi-scale detection framework called DFCformer based on the transformer. DFCformer is mainly composed of DMViT (Depatch Multi-scale Vision Transformer), FRGC (Feature Reconciliation Guidance Component), and CAIM (Cascaded Attentional Interactive Module). DMViT is the backbone network, which abandons the fixed-size patch embedding in the PVT [29] and introduces the DePatch embedding [30] to mitigate the semantic damage caused by image splitting and retain the complete semantic information in a patch. Specifically, DMViT adopts a scale adaptive attention mechanism to enhance the ability of different attention heads at the same layer to model objects with different scales, improve the ability to save fine-grained details, and reduce the computational power consumption. Meanwhile, FRGC helps to break the barriers between feature layers, filter the interference information on different feature layers, and improve the purity of crucial details. Besides, to overcome the limitation of category confusion on network accuracy, CAIM is employed to better integrate features with inconsistent semantics and scales and improve the ability of network fine-grained feature interpretation. The main contributions of this study are summarized as follows:

- A backbone network combining deformable patch embedding and a multi-scale visual converter is proposed to improve the ability to capture the details of multi-scale aerial targets in complex scenes.
- The cross-layer feature reconciliation guidance component enhances the semantic information of crucial features on different layers, which helps to alleviate category confusion and realize accurate classification and regression.
- The attention-based feature fusion mechanism strengthens the information integration between multi-scale and multi-semantic features and overcomes the limitation of category confusion on network accuracy.

## 2. Related Work

Relevant prior work includes ConvNets for object detection, vision transformer for object detection, and aerial object detection.

### 2.1. ConvNets for Object Detection

As a mainstream and standard deep learning model, ConvNets is often used in object detection. According to the use of the region-of-interest proposal step, the existing ConvNets models can be divided into two categories: two-stage models and one-stage models. The two-stage models [14,31] represented by R-CNN [32] achieve good performances. Following the single-stage structural design strategy, the Yolo series [33–35] and SSD [36] have attracted wide attention because of their simple network structure and high reasoning performance.

The remarkable achievements of these models based on ConvNets in general object detection tasks inspire us to extend object detection to aerial object detection tasks, promote the intellectual development of aerial scene perception, and expand the scope of engineering applications of computer vision.

### 2.2. Vision Transformer for Object Detection

The multi-head self-attention mechanism is the core of the transformer, which enables the transformer to learn the complex dependencies from sequence to sequence [37]. Vision transformer [22] (ViT) divides the image into non-overlapping image patch sequences, thus producing a novel image classification model and creating a precedent for the transformer to migrate to object detection tasks. Since the pioneering ViT model was proposed, researchers have designed many excellent models by optimizing the model structure and the component's function. For example, DETR [38] abstracts objects detection as a prediction model of transformer and loss function, thus eliminating the dependence on handcrafted modules and operations (i.e., RPN and NMS). To overcome the limitation of the slow convergence speed of DETR, Zhu et al. [39] proposed Deformable DETR. Deformable DETR adopts a deformable attention mechanism, which improves the aggregation of cross-scale feature

maps under controllable computational cost and keeps the balance between performance and inference rate. Following the attention optimization strategy, Swin [40] and PVT [29] establish flexible global attention to effectively reduce the computational complexity of the model and improve the detection performance. Meanwhile, a plethora of studies on optimizing the patch embedding has made exemplary achievements. T2T-ViT [41] processes tokens in the way of aggregation and recursion to ensure the integrity expression of local information. MPViT [42] builds multi-path patch embedding to realize fine feature and coarse feature representation at the same feature level, which helps to detect dense objects. Considering the different information expressed by varying levels of features, LVT [43] introduces Convolutional Self-Attention (CSA) to deal with the underlying features by including dynamic kernels and learnable filters. The introduction of Recursive Atrous Self-Attention (RASA) is conducive to extracting multi-scale context information and increasing the representation ability of marginal additional parameter cost. DeepViT [44] proposes a mechanism to regenerate attention, which increases the diversity of attention maps at different layers to control the computational cost and alleviate the performance saturation problem caused by network deepening. RVT [45] introduces position-aware attention scaling mechanism and patch-wise enhancement mechanism to improve the robustness and generalization ability of the transformer-based network. These excellent studies enhance the expansion of the transformer and promote us adopt the transformer in object detection.

### 2.3. Aerial Object Detection

With the release of large annotation datasets for aerial object detection, many researchers attempt to apply deep learning models to aerial object detection. Pang et al. [46] proposed an end-to-end detection framework $R^2$-CNN to improve the reasoning speed and reduce severe false alarms in aerial object detection. Then, Pan et al. [47] enhanced the generalization training process of the model based on maximizing the alignment of neurons and improved the detection performance for dense objects in aerial scenes. Ma et al. [48] employed the deep separable convolution in the transformer, which significantly reduces the memory and computation cost of multi-scale features. Ran et al. [49] maintained the small-scale object in the down sampling operation without losing features, and they adopted the strategy of enhancing the attention of the feature image channel and fusing the multi-scale context information of the network. Xu et al. [50] combined reinforcement learning and strategy gradient to build a scale zoom detection network called AdaZoom, which improves the robustness of the network to multi-scale aerial objects. These methods provide significant reference values for the future study of aerial object detection. However, how to overcome the challenges of aerial object detection still needs further investigation.

### 3. Methods

This section mainly describes the structure and action mechanism of the three main components in DFCformer.

Based on the above analysis and comparison, this paper adopts a four-stage progressive feature fusion detection strategy and proposes a multi-scale detection architecture DFCformer based on the transformer structure to overcome the challenges, (i.e., large-scale variation and confused distinguishing features) in aerial object detection. As shown in Figure 3, DFCformer is divided into three parts. The first part takes DMViT as the backbone and integrates deformation patch embedding and multi-scale adaptive self-attention to improve the ability to capture features of aerial objects. See Section 3.1 for the details of DMViT. The second part is the FRGC, which guides the feature interaction at different layers to handle the significant difference between aerial object classes and high similarity within classes and promotes the fine-grained detection of the framework. The discussion of FRGC is presented in Section 3.2. The third part is the CAIM, which adopts the cascade attentional fusion strategy to conduct hierarchical reasoning on the relationship between different levels of features. Also, CAIM mines the rich complementary information in multi-scale

features and enhances the overall performance of the framework. The description of the CAIM is presented in Section 3.3.

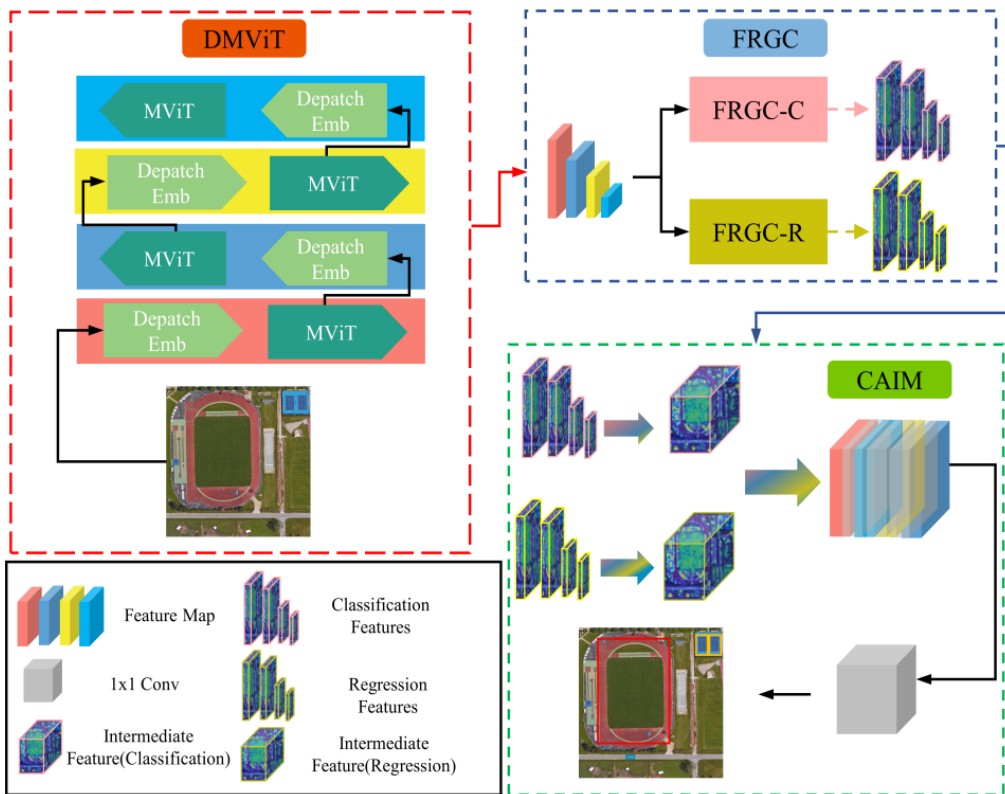

**Figure 3.** The structure of the DFCformer. The DMViT consists of deformable patch embedding and a multi-scale visual transformer. FRGC-C and FRGC-R are the feature processing units of the classification task and regression task in FRGC, respectively. CAIM is a cascade attention interaction module to enhance the modeling and reasoning of the relationship between different scale features.

### 3.1. DMViT

To improve semantic recognition ability and reduce category confusion, this paper collects and correlates scene information from more prominent neighborhoods to infer the correlation between objects, which is the key to capturing feature information from aerial images of a comprehensive visual range of complex scenes. However, for traditional ConvNets, the local filtering of the convolution kernel limits its ability to obtain global context information. In contrast, the transformer can construct the dependency between image feature blocks globally and retain sufficient spatial information for object detection through self-attention.

A series of detection frameworks have been proposed based on transformer structure. However, these detection frameworks still have limitations in accurate detection of multi-scale aerial objects with a large field of view and complex background, and this is mainly reflected in the following two aspects: (1) Figure 4a shows that using fixed-size patch embedding to segment images cannot accurately capture the critical information of objects of different scales and the consistent characteristics of the same object under different geometric changes. (2) Figure 4b shows that the model largely ignores the multi-scale of objects in the attention layer. The potential attention mechanism of the model only depends on the marked static receptive field and the unified information granularity in an attention layer, so it cannot capture the features of different scales simultaneously.

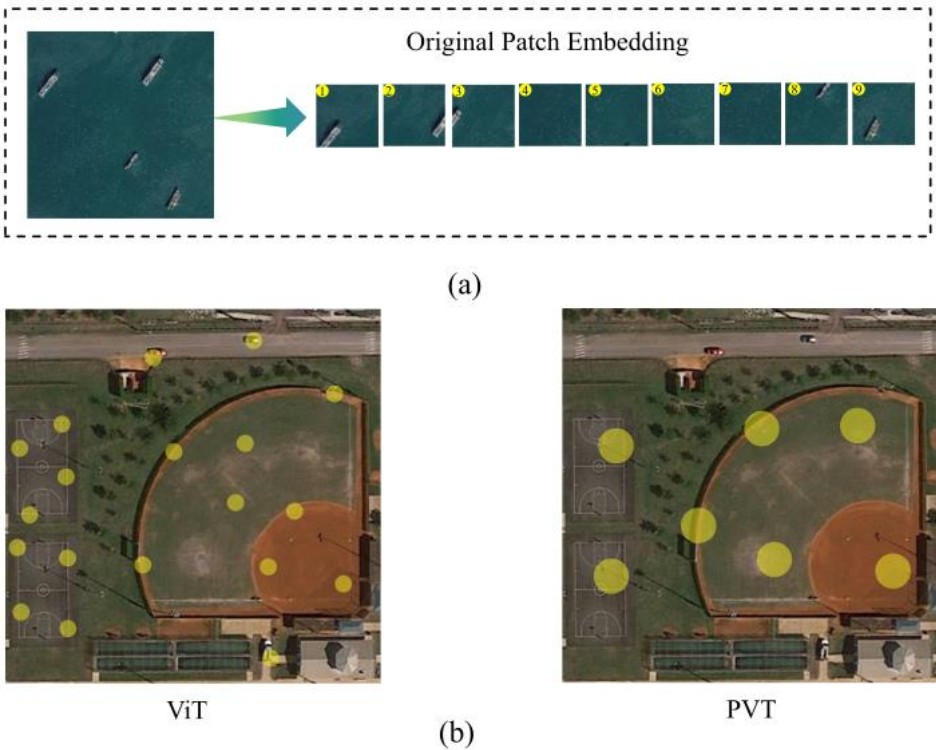

**Figure 4.** The limitations of the transformer-based networks in aerial object detection. (**a**) The original patch embedding module divides the image in a fixed way. It sometimes destroys the semantics of objects. (**b**) The size of the circle represents the receptive field size of the token, and the number of circles represents the number of tokens in the self-attention calculation, both of which reflect the computation cost. ViT focuses on fine-grained objects, but it has an extremely heavy computation cost. PVT reduces the computation cost, but it only focuses on coarse-grained large objects and ignores fine-grained objects.

To overcome this limitation, this paper proposes a new construction called MViT (Multi-scale vision transformer), as shown in Figure 5. Different attention heads at the same layer of MViT can effectively model objects of various sizes and explain coarse-grained and fine-grained characteristics. Also, MViT has good computation efficiency and can retain fine-grained details and obtain more discriminate information.

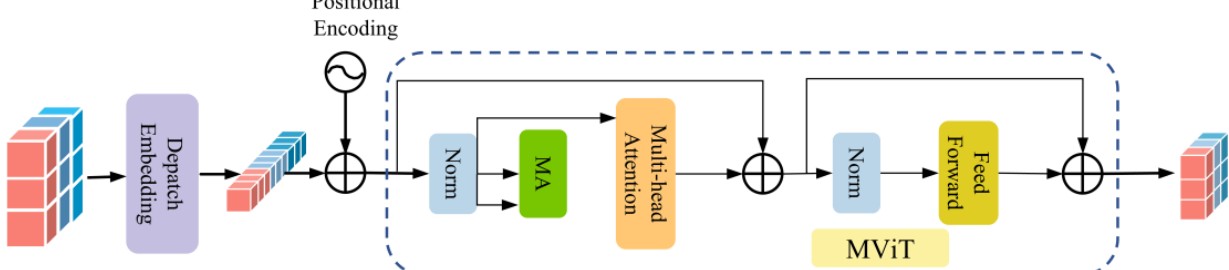

**Figure 5.** The details of MViT, MA (Multi-scale aggregation) operations of the tokens enhance the multi-scale adaptability of MViT to capture multi-scale features.

Transformer originates from natural language processing, and its core task is to deal with the mapping relationship between sequences. DePatch (Deformable Patch Embedding) relieves the constraints in the original patch embedding and endows patch embedding with deformable ability, thus better locating essential structures and retaining rich semantic information. The details of Depatch are illustrated in Figure 6.

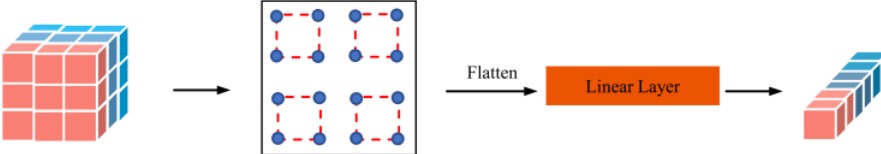

(a) Orignal patch embedding in PVT

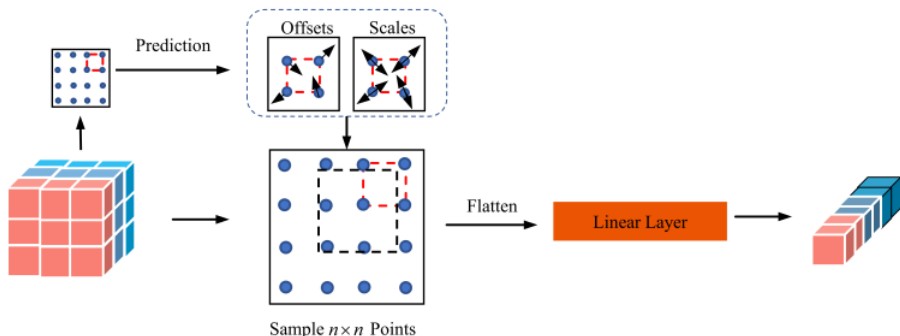

(b) Depatch embedding $(n = 3)$

**Figure 6.** Instruction of the depatch embedding.

Given an input image $X \in \mathbb{R}^{H \times W \times C}$, DePatch divides $X$ into $N$ patches with a size of $t \times t$ ($t = \lceil H \times W/N \rceil$), and the sequence of $N$ patches is donated as $\{s_m | 1 \leq m \leq N\}$. This paper defines the DePatch to better explain the process. First, the patch $s_m$ in the input image is approximated to a rectangular region. The center coordinate of the rectangular region is $(x_c^m, y_c^m)$, and its left-top and right-bottom corner coordinates are $(x_l^m, y_l^m)$ and $(x_r^m, y_r^m)$, respectively. The mathematical derivation is shown as follows:

$$x_l^m = x_c^m + \alpha x^m - \frac{t_w^m}{2} \tag{1}$$

$$y_l^m = y_c^m + \alpha y^m - \frac{t_h^m}{2} \tag{2}$$

$$x_r^m = x_c^m + \alpha x^m + \frac{t_w^m}{2} \tag{3}$$

$$y_r^m = y_c^m + \alpha y^m + \frac{t_h^m}{2} \tag{4}$$

Specifically, $(\alpha x, \alpha y)$ is the predicted offset; $t_w^m$ and $t_h^m$ are the scale of the patch $s_m$.

This paper predicts $(\alpha x^m, \alpha y^m, t_w^m, t_h^m)$ for all patches and then embeds them with the rectangular region. The offset and scale can be shown as follows:

$$\alpha x, \alpha y = Tanh\left(W_{offset} \cdot L(X)\right) \tag{5}$$

$$t_w, t_h = ReLU(Tanh(W_{scale} \cdot L(A) + b_{scale})) \tag{6}$$

where $L(\cdot)$ is a single linear layer; $W_{offset}$ and $W_{scale}$ are adopted to predict the offset and scale, Considering that the raw image contains little semantic information, the first module is difficult to predict the offset and scale beyond its region, so these weights (offset and scale) are initialized to zero at the beginning of training; $b_{scale}$ corrects the deviation of the initial state to ensure that each patch focuses on the same rectangular area. Considering that the size of the generated region is different and the predicted coordinate forms are diverse, the computational power consumption is increased. In this paper, interpolation sampling is adopted to simplify the calculation. By taking $n \times n$ sampling points evenly

in the region, each sampling coordinate is $\left(p_x^i, p_y^i\right)$ $(1 \leq i \leq n \times n)$, and the features of all sampled points $\{p^i\}_{1 \leq i \leq n \times n}$ are flattened to generate DePatch, as shown in Equation (7).

$$s^m = W_{DePatch} \cdot concat\left(p^1, p^2, \ldots, p^{n \times n}\right) + b_{DePatch} \tag{7}$$

The feature at each sampling point is calculated by bilinear interpolation in Equation (8).

$$X(p_x, p_y) = \sum_{q_x, q_y} G(p_x, p_y; q_x, q_y) \cdot X(q_x, q_y) \tag{8}$$

$$G(p_x, p_y; q_x, q_y) = max(0, 1 - |p_x - q_x|) \cdot max(1, 1 - |p_y - q_y|) \tag{9}$$

After the above operation and processing, the constraint of the fixed scale of patch size can be released. Meanwhile, the location and scale of each patch can better adapt to the distribution of instances and reduce the semantic damage caused by image segmentation.

The flexible segmentation style retains rich semantic information. To improve the perception ability to the feature information of multi-scale objects, inspired by reference [51], this paper proposes MSSA (multi-scale self-attention) to replace the vanilla multi-head self-attention in the transformer to obtain MViT (multi-scale vision transformer).

MSSA divides the self-attention into several subsets with different attention granularities, realizes scale variations, and enhances the modeling ability for multi-scale objects. Specifically, in the fine-grained subset, MSSA focuses on the expression of local detail information and weakens the modeling of long-distance context information. In the coarse-grained subset, MSSA efficiently aggregates global information, focuses on sensing long-distance up and down information, fully captures features, and reduces the computational power consumption. MSSA first projects the input sequence $A \in \mathbb{R}^{H \times W \times C}$ (generated by DePatch) into $Q$ (query), $K$ (key), and $V$ (value). As shown in Figure 7, unlike the prior vision transformer, the scales of $K$ and $V$ are sorted according to a certain proportion:

$$Q_c = XW_c^Q \tag{10}$$

$$K_c = MA(X, r_c)W_c^K \tag{11}$$

$$V_c = MA(X, r_c)W_c^V \tag{12}$$

where $W_c^Q$, $W_c^K$, and $W_c^V$ are the linear projection parameters in the $c$-th attention head, $MA(\cdot)$ denotes the multi-scale aggregation of the tokens, and $r_c$ is the adjustment of the rate in the $c$-th attention head.

Then, the multi-scale self-attention is calculated in Equation (13)

$$MSSA_c = \text{Softmax}\left(\frac{Q_c K_c^T}{\sqrt{d_m}}\right) \tag{13}$$

where $\sqrt{d_m}$ is the dimension.

The multi-scale of $K$ and $V$ is conducive to capturing multi-scale features. However, the proportion $r_c$ increases the computational power consumption. When $r_c$ is too large, K and V become smaller, and the computational power consumption decreases, but the feature perception ability is weakened. On the contrary, when $r_c$ is too small, K and V become larger, and the feature perception ability is enhanced, but the computational power consumption increases suddenly. Therefore, the value of $r_c$ should be set under the balance of computational power consumption and model performance.

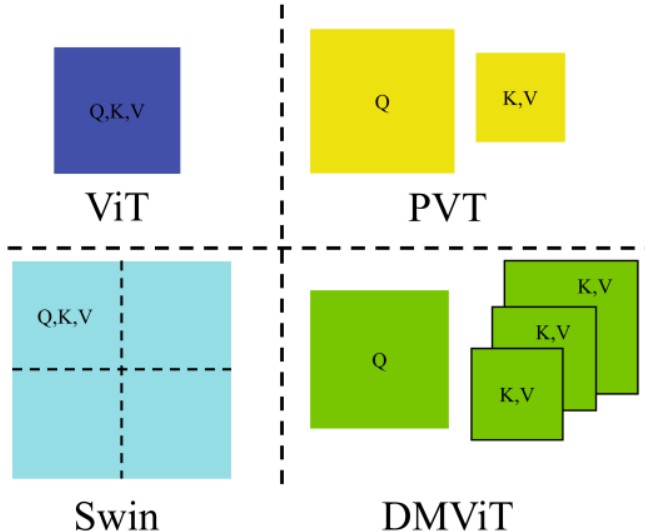

**Figure 7.** Comparisons with different self-attention mechanisms in ViT, PVT, Swin, and DMViT. The global self-attention ViT focuses on small-scale feature maps. Swin focuses on the local region of large-scale feature maps. In the self-attention of PVT, Q focuses on the global while K and V focus on the local. Unlike the first three self-attention mechanisms, DMViT introduces multi-scale token aggregation in the self-attention to obtain the keys and values of different scales.

### 3.2. FRGC

The complex and changeable distribution characteristics of ground objects and the unpredictable environmental interference in the field of view make it more difficult to interpret aerial scenes. Based on the analysis of the overall operation of object detection, this paper proposes an efficient feature extractor DMViT, which only captures external information. To filter redundant interference information, this paper designs FRGC to optimize the information processing process and alleviate category confusion. The details of FRGC are presented in Figure 8.

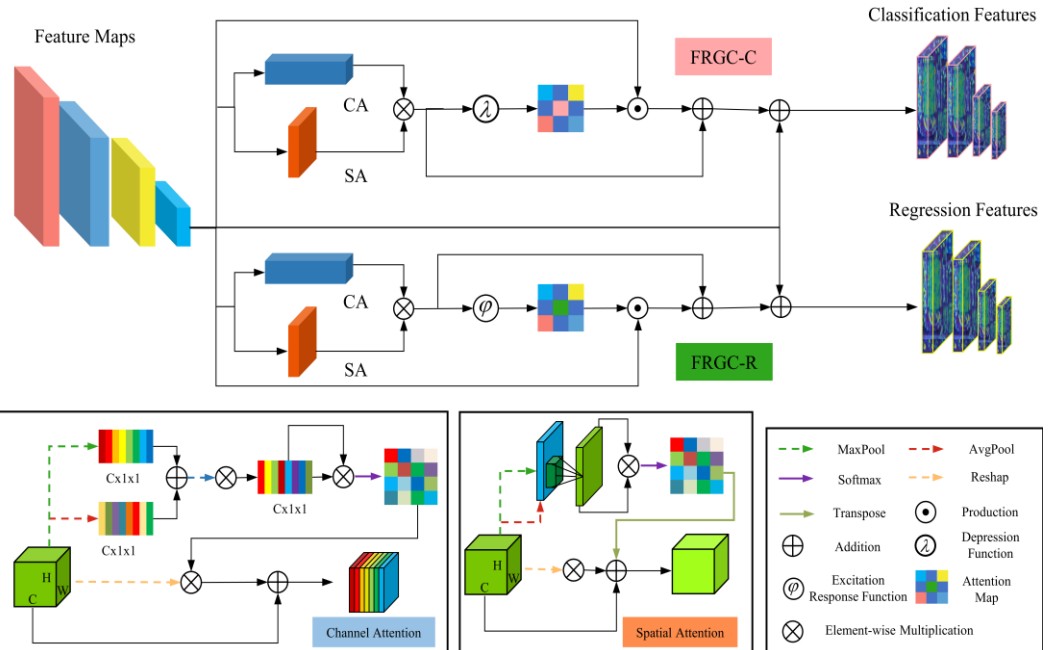

**Figure 8.** Illustration of the FRGC. FRGC-C and FRGC-R are the feature processing units of the classification task and regression task in FRGC, respectively.

Considering the different proportions of information carried by different feature layers and the different critical features required by object detection tasks (a large proportion of low-level feature texture information is conducive to regression subtasks, while a large proportion of high-level feature semantic information is conducive to classification subtasks), the FRGC adopts a guiding cross-layer interaction mechanism to break the barriers between feature layers, filters the interference information on different levels of features, and improves the information discrimination and processing ability of multi-scale critical features of components.

In Figure 8, FRGC is composed of multiple FRGC-R and FRGC-C units. It refines the semantic perception and texture expression of aerial objects layer by layer, efficiently interacts with different semantic information and texture information, and generates the critical features required by regression and classification subtasks. Specifically, in FRGC-R, the texture features suppress the background noise and help to extract the critical features required by regression from the detailed information of the object. Meanwhile, in FRGC-C, the semantic guiding features effectively suppress the background noise, restrict the diffusion of semantic features, and help to obtain more classification features.

FRGC-R operates in two steps: first, the features $f_i$ from DMViT are interacted to suppress the background noise and improve the proportion of regression information in the enhanced features $f_i'$.

$$R = A_c(f_i) \otimes A_s(f_i) \tag{14}$$

$$f_i' = R + \lambda_{reg}(\sigma(R)) \odot f_i + f_i \tag{15}$$

$$\lambda_{reg}(x) = \begin{cases} x & x \leq 0.5 \\ 1-x & \text{else} \end{cases} \tag{16}$$

where $f_i'$ denotes the enhanced features; $A_c$ denotes the channel-wise attention operation; $A_s$ denotes the spatial attention operation; $\odot$ denotes the tensor product; $\otimes$ denotes element-wise multiplication; $\sigma$ denotes the sigmoid function; $\lambda_{reg}(\cdot)$ denotes the depression function, and it mainly suppresses the regions with high response in the regression features to promote the model to find potential visual clues and realize accurate positioning.

Then, FRGC-R guides the enhanced features for interactive perception and generates the critical features $f_i^r$ required by the regression subtask.

$$f_1 = C(C(f_i)) \odot C(f_{i+1}) \tag{17}$$

$$f_i^r = C(C(f_1) + C(f_i)) \tag{18}$$

where $f_1$ denotes the intermediate variable, and $C(\cdot)$ denotes the CBR operation (Convolution, batch normalization, and ReLu).

FRGC-C operates also in two steps: the features $f_i$ from DMViT are guided to suppress background noise and improve the proportion of classification information in the enhanced features $f_i''$.

$$R = A_c(f_i) \otimes A_s(f_i) \tag{19}$$

$$f_i'' = f_i + R + \varphi_{cls}(\sigma(R)) \odot f_i \tag{20}$$

$$\varphi_{cls} = \frac{1}{1 + e^{-\theta(x-0.5)}} \tag{21}$$

where $f_i''$ denotes the enhanced features; $\varphi_{cls}(\cdot)$ denotes the excitation response function, and it mainly focuses on the high response part of the feature map and filters out the positioning clues used for interference noise or regression; $\theta$ denotes a factor used to regulate the intensity of feature activation (it is set to 18 in our experiment). Because the regional response of some critical features can stimulate the role of high-response key classification features and realize accurate classification, the irrelevant features with an attention weight of less than 0.5 cannot reach the activation threshold and are suppressed.

In this way, the interference of irrelevant classification areas can be eliminated, and the fitting performance and anti-misjudgment mechanism of the model are improved.

Then, FRGC-C guides the enhanced features for interactive perception and generates the critical features $f_i^c$ required by the regression subtask.

$$f_2 = C(C(f_i)) \odot C(f_{i+1}) \tag{22}$$

$$f_i^c = C(C(f_2) + C(f_i)) \tag{23}$$

where $f_2$ denotes the intermediate variable, and $C(\cdot)$ denotes the CBR operation (Convolution, batch normalization, and ReLu).

### 3.3. CAIM

In Sections 3.2 and 3.3, the efficient multi-scale feature information detector and the powerful critical feature processing operations are introduced, respectively. This section focuses on organically integrating the features at different levels to obtain the final prediction result.

Feature fusion is an unreachable part of the object detection network, which combines features from different layers or branches: (1) The information in low layers can be further enhanced by feature fusion; (2) The middle layers consider both semantic information and detail information and can adaptively adjust the proportion of different abstract information to realize better utilization of flexible features; (3) At the top layers, richer semantic information can be mined when considering the adjacent resolution.

For aerial object detection, to overcome these challenges (i.e., large-scale variations and confused distinguishing features), an intuitive method is adopted to establish a multi-stage detection framework and fuse multi-scale features for object prediction [52]. However, the feature fusion adopted in most aerial object detection networks [53–55] ignores the modeling and reasoning of the relationship between different scale features, which is not conducive to the localization of aerial objects and the mining of object features.

The attention mechanism imitates the human's cognitive awareness of specific information. It enlarges essential details and pays more attention to vital aspects. In recent years, the attention mechanism has widespread use in visual recognition. The advantages of the attention mechanism in long-distance context modeling contribute to the interactive clustering of global features. Inspired by the study [56], this paper designs a cascade attention interaction module (CAIM) to enhance the modeling and reasoning of the relationship between different scale features.

The core idea of CAIM is that by changing the size of the spatial pool, channel attention can be realized on multiple scales. Meanwhile, considering the emphasis on different levels of feature expression and computational power consumption, CAIM adopts a cascade way to integrate multi-scale features step by step, as shown in Figure 9.

Figure 9 demonstrates the details of CAIM.

First, the features $f_i^r$ and $f_{i+1}^r$ from FRGC-R are processed by broadcast addition to obtain the intermediate feature $\overline{f}^r$. Then the intermediate feature $\overline{f}^r$ is input into a feature aggregation component to realize point-to-point channel interaction of spatial location.

$$\overline{f}^r = \sigma(B(PC(\gamma(B(PC(f^r)))))) \tag{24}$$

where $B(\cdot)$ is denotes batch normalization, $PC(\cdot)$ denotes the point-wise convolution, $\gamma(\cdot)$ denotes the ReLu operation, and $\sigma$ denotes the sigmoid function.

$$f_l^r = \alpha f_i^r \otimes \overline{f}^r + (1 - \alpha) f_{i+1}^r \otimes \overline{f}^r \tag{25}$$

where $\otimes$ denotes the element-wise multiplication, and $\alpha$ is the adjustment coefficient used to adjust the weight of features of different layers in combination features.

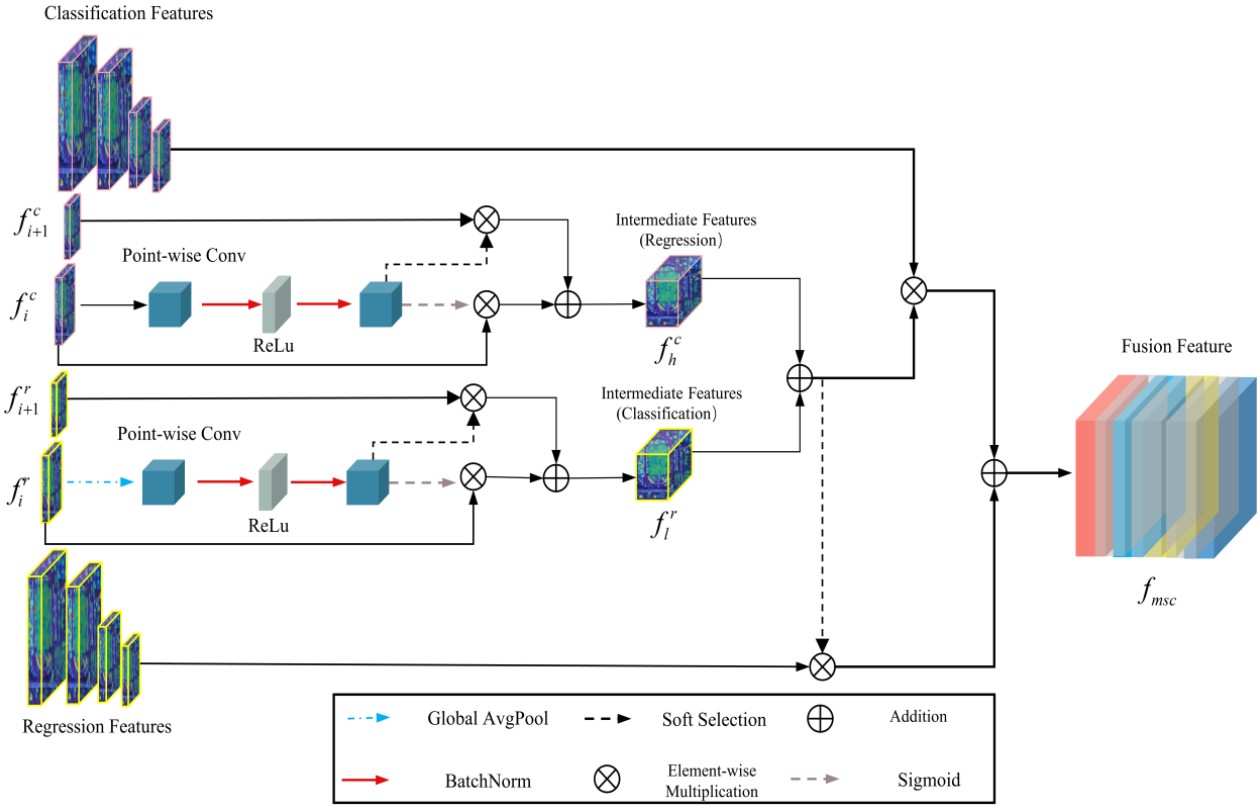

**Figure 9.** Illustration of the proposed CAIM, which is a cascade attention interaction module to enhance the modeling and reasoning of the relationship between different scale features.

Similarly, the features $f_i^c$ and $f_{i+1}^c$ are fused in the aggregation process.

Firstly, the features $f_i^c$ and $f_{i+1}^c$ from FRGC-C are processed by broadcast addition to obtain the intermediate feature $\overline{f}^c$. Then, the combined feature $\overline{f}^c$ is input into a feature aggregation component to realize the point-to-point channel interaction of spatial location based on global average pooling operation.

$$\overline{f}^c = \sigma(\text{PC}(\gamma(\text{B}(\text{PC}(\text{GP}(f^c)))))) \tag{26}$$

where $\text{GP}(\cdot)$ denotes the global average pooling operation.

Finally, the intermediate feature $\overline{f}^c$ is combined with the original feature to obtain the classification combined feature $f_h^c$.

$$f_h^c = \beta f_i^c \otimes \overline{f}^c + (1 - \beta) f_{i+1}^c \otimes \overline{f}^c \tag{27}$$

where $\beta$ is the adjustment coefficient used to adjust the weight of features of different layers in combination features.

Based on the above fusion branch interaction, the combined features $f_l^r$ and $f_h^c$ are processed in a similar way to obtain the final multi-scale fusion feature $f_{msc}$.

$$f_{msc} = \delta f_l^r \otimes \overline{f} + (1 - \delta) f_h^c \otimes \overline{f} \tag{28}$$

$$\overline{f} = \sigma(f_l^r + f_h^c) \tag{29}$$

where $\overline{f}$ is the combined feature of $f_l^r$ and $f_h^c$, and $\delta$ is the adjustment coefficient used to adjust the weight of features of different layers in the final multi-scale fusion feature.

### 3.4. Network Configuration

Several essential components of DFCformer are introduced in detail in Sections 3.1–3.3 above. Designing an efficient network based on these three components is the core content of our discussion in this section. From the perspective of system theory, the whole is greater than the sum of parts, and each component should be organically connected to form a reasonable structure to maximize the overall performance. An appropriate design for a deep learning network is not the simple stacking of various elements but finding a balance between performance and computational power consumption [57].

The core component of the DFCformer is the backbone, and the remaining two parts are configured according to the structural characteristics of the backbone. Meanwhile, the DMViT in each stage of the backbone is composed of one Depatch and several MViTs. Therefore, this paper mainly analyzes the configuration of MViT in DMViT in different stages. Inspired by the reference [40], this paper focuses on comparing the DMViT under three configurations, as shown in Table 1.

**Table 1.** Different configurations of DMViT. *C* is the dimension, *N* denotes the number of MViT, *head* represents the number of heads, and $r_c$ is the adjustment of the rate in the *c*-th attention head.

| | DMViT-S | DMViT-M | DMViT-L |
|---|---|---|---|
| | Depatch Embedding | | |
| Stage1 | $r_c = 8 \quad N_1 = 1$ $C_1 = 64 \quad head = 2$ | $r_c = 8 \quad N_1 = 1$ $C_1 = 64 \quad head = 2$ | $r_c = 8 \quad N_1 = 1$ $C_1 = 64 \quad head = 2$ |
| | Depatch Embedding | | |
| Stage2 | $r_c = \begin{cases} 2 & c < \frac{head}{2} \\ 4 & c \geq \frac{head}{2} \end{cases}$ $C_2 = 128 \quad N_2 = 2 \quad head = 4$ | $r_c = \begin{cases} 2 & c < \frac{head}{2} \\ 4 & c \geq \frac{head}{2} \end{cases}$ $C_2 = 128 \quad N_2 = 3 \quad head = 4$ | $r_c = \begin{cases} 2 & c < \frac{head}{2} \\ 4 & c \geq \frac{head}{2} \end{cases}$ $C_2 = 128 \quad N_2 = 3 \quad head = 4$ |
| | Depatch Embedding | | |
| Stage3 | $r_c = \begin{cases} 1 & c < \frac{head}{2} \\ 2 & c \geq \frac{head}{2} \end{cases}$ $C_2 = 128 \quad N_3 = 4 \quad head = 8$ | $r_c = \begin{cases} 1 & c < \frac{head}{2} \\ 2 & c \geq \frac{head}{2} \end{cases}$ $C_2 = 128 \quad N_3 = 6 \quad head = 8$ | $r_c = \begin{cases} 1 & c < \frac{head}{2} \\ 2 & c \geq \frac{head}{2} \end{cases}$ $C_2 = 128 \quad N_3 = 8 \quad head = 8$ |
| | Depatch Embedding | | |
| Stage4 | $r_c = 1 \quad N_1 = 1$ $C_1 = 512 \quad head = 16$ | $r_c = 1 \quad N_1 = 1$ $C_1 = 512 \quad head = 16$ | $r_c = 1 \quad N_1 = 1$ $C_1 = 512 \quad head = 16$ |

In addition, this paper selects Top-1 accuracy to measure the detection accuracy of the three variants, i.e., DMViT-S, DMViT-M, and DMViT-L, and selects flops to measure the computational cost of the variants on the ImageNet-1k [58], as shown in Figure 10. Overall, the three variants perform well, and the worst accuracy is more than 77%, indicating the excellent detection performance of DMViT. Specifically, DMViT-S has the simplest structure and the lowest computational cost, but its accuracy is the lowest. On the contrary, DMViT-L achieves the highest accuracy of 82.2%, but its structure is the most complex, and the computational cost is 8.3 G FLOPs, which is 2 times that of DMViT-S. Compared with the DMViT-S, DMViT-M has a more complex structure and higher computational cost, but its accuracy is increased by 2.9%. Meanwhile, compared with DMViT-L, the accuracy of DMViT-M is decreased by 0.5%, but its computational power consumption is reduced by 22%. To sum up, DMViT-M achieves good performance under an acceptable structural complexity and computational cost. Therefore, this paper adopts DMViT-M as the backbone.

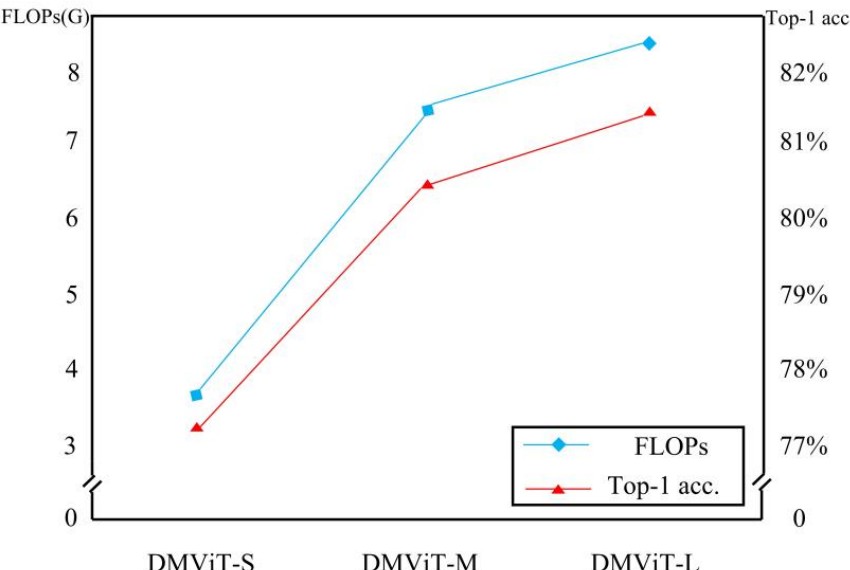

**Figure 10.** Quantitative comparisons among the variants.

*3.5. Multi-Task Learning*

Aerial object detection is a multi-task learning problem, including regression and classification subtasks. The classification subtask is to learn the identification features of the key or prominent parts of the object [59]. The regression subtask is to locate the whole object and its boundary accurately. This paper further proposes multi-task learning, which includes (1) screening high-quality anchors, and (2) optimizing the calculation mode of the branch task loss.

Considering that the classification score and the IoU between the prediction box and the ground truth are the evaluation indicators of the two subtasks, this paper uses the high-order combination of classification score and IoU to screen the anchor box. Specifically, the following indicators are designed to calculate the quality of the anchors to encourage the network to dynamically focus on high-quality anchors from the perspective of joint optimization.

$$t = s^\eta \times o^\tau \tag{30}$$

where $\eta$ and $\tau$ are used to adjust the influence of anchors in the classification score ($s$) and IoU value ($o$), respectively.

To improve the classification score of high-quality anchors and reduce the score of low-quality homogeneous anchors, this paper employs the maximum IoU value ($o$) in each instance corresponding to the maximum value of $t$. The binary cross-entropy (BCE) on the positive anchors for the classification task is defined as:

$$L_{bce} = \sum_{i=1}^{N_{pos}} BCE(s_i, t_{\max}) \tag{31}$$

where $i$ is the $i$-th positive anchor in an instance, and $t_{\max}$ is the maximum value of $t$.

The final classification task loss function is defined as:

$$L_{cls} = \sum_{i=1}^{N_{pos}} |s_i - t_{\max}|^\omega \ BCE(s_i, t_{\max}) + \sum_{j=1}^{N_{neg}} s_j^\omega BCE(s_j, 0) \tag{32}$$

where $\omega$ is the scale parameter, and $j$ is the $j$-th negative anchor in an instance.

Similarly, to increase the proportion of high-quality anchor boxes in the regression task and reduce the proportion of low-quality anchor boxes in the regression task, this paper recalculates the GIoU loss [60] on each anchor box based on $t_{\max}$.

$$L_{reg} = \sum_{i=1}^{N_{pos}} t_{\max} L_{GIoU}(b_i, g_i) \tag{33}$$

where $g_i$ and $b_i$ denote the ground-truth box and the predicted bounding box, respectively.

The total loss is the sum of $L_{cls}$ and $L_{reg}$:

$$L = v L_{cls} + \varsigma L_{reg} \tag{34}$$

where $v$ and $\varsigma$ are the focusing parameters.

## 4. Experiment

### 4.1. Dataset

Object detection is a data-driven application in computer vision. The performance of the deep learning network depends on the quality and quantity of the given data. Different from raw image data sets such as ImageNet [58] and MSCOCO [61], a challenging and excellent aerial object detection data set should have the following properties:

- The scale of the dataset is huge. With the increasing demand for aerial object detection, the detector needs higher critical generalization. Many algorithms have been proposed and perform well on small datasets, but their performance decreases rapidly on large datasets. Therefore, to train the detector more comprehensively, the corresponding dataset needs a large volume of object instances and images.
- Data samples are rich in detail. The similarity within the aerial class increases the demand for fine-grained recognition. An excellent detection algorithm should be able to correctly identify objects belonging to specific subcategories. Most of the existing datasets contain coarse-grained information and lack detailed information, which makes it difficult to improve the detection performance of deep learning methods.
- The image quality of data samples is high. Factors such as rain, fog, cloud, and jitter may interfere with the quality of aerial images. It is challenging to train excellent algorithms with low-quality samples to meet the requirements of aerial object detection.

Based on the above analysis, this paper comprehensively evaluates the standard aerial object datasets (DIOR [26], DOTA [62], VisDrone [63], VHR-10 [25], FAIR1M [64], RSOD [65], VEDAI [66], and HRSC2016 [67]) in terms of sample size (Images), the number of instances (Instances), type distribution (Categories), and sample quality (Quality) to select the best dataset. The results presented in Figure 11 show that these datasets perform differently in the four indicators, and FAIR1M has more significant advantages in all-around performance than other datasets. Thus, this paper selects FAIR1M as the data source.

FAIR1M is a new benchmark dataset for fine-grained object detection in high-resolution aerial scenes. Specifically, the training set and test set of the FAIR1M contain 16,488 images and 8137 images, respectively. The overall data set is composed of 5 categories and 37 subcategories. All classification and example statistics are illustrated in Figure 12.

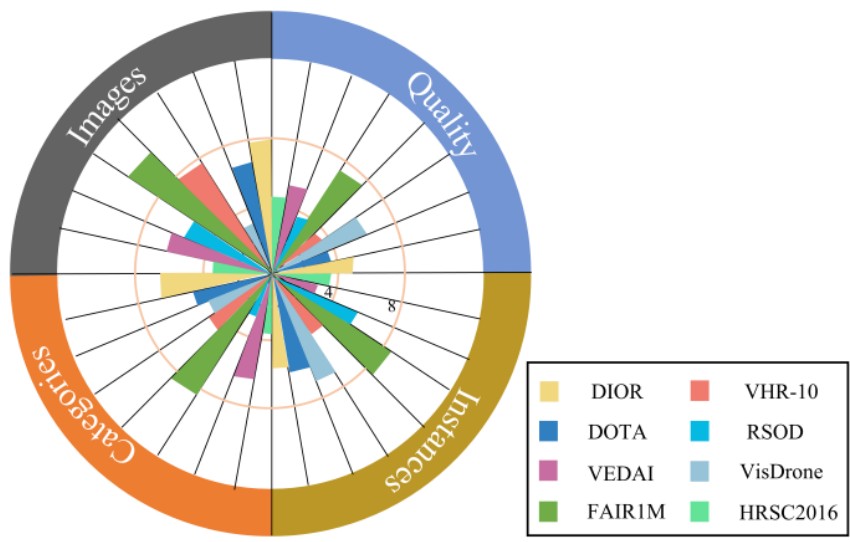

**Figure 11.** Multi-dimensional comparisons of typical datasets in the field of aerial object detection.

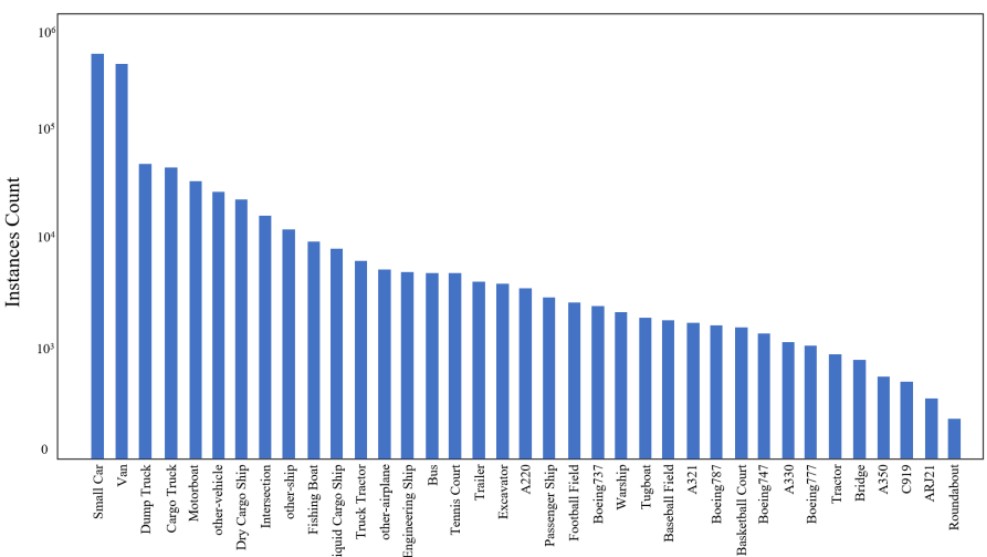

**Figure 12.** The specific instance distribution of dataset categories.

*4.2. Evaluations Metrics*

In object detection, $IoU$ is the most widely used index to measure the overlap between the prediction box (P) and the ground-truth box (G). However, the index that measures only from the geometric dimension cannot well reflect the quality of the information extracted by the prediction box. As shown in Figure 13, the $IoUs$ of (a) and (b) are the same, but there is more interference in the information extracted by the prediction frame of a, which is not conducive to the identification of the instance category of the scene.

Referring to conclusions from the study [67], this paper selects $FIoU$ as the evaluation index to penalize the exceptional results. Similar to the definition of $IoU$, $TP$, $FP$, $TN$, $FN$, $AP_F$, and $mAP_F$ are defined for $FIoU$.

$$FIoU = \sqrt[3]{\frac{P \cap G}{G \cup P} \cdot \frac{P \cap G}{G} \cdot \frac{P \cap G}{P}} \tag{35}$$

$$\text{Precision}_F = \frac{FIoU \cdot TP \cdot score_{TP}}{TP \cdot score_{TP} + FP \cdot score_{FP}} \tag{36}$$

$$\text{Recall}_{\text{F}} = \frac{FIoU \cdot TP}{TP + FN} \tag{37}$$

where *score* denotes the classification score of the prediction box.

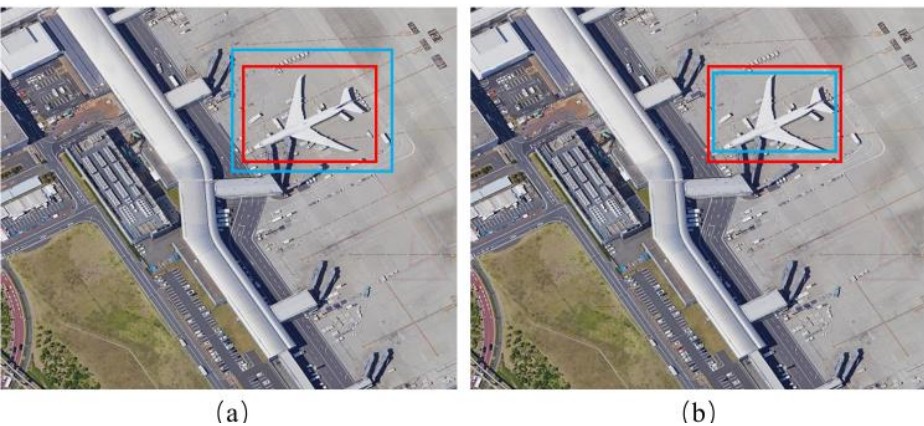

(a)           (b)

**Figure 13.** Detection results in two samples (Red: ground truth, Blue: prediction). (**a**,**b**) are the same in IoU, but the predicted bounding box in (**b**) contains less confusing information.

*4.3. Implementation Details*

To better fit the limited computing power of the drone platform, the computer equipped with NVIDIA rtx3090 GPU is selected as the experimental platform. Following the setting of the Swin Transformer [40], this paper uses ImageNet-1k to pre-train all transformer-based backbones and fine-tunes them on the training set of FAIR1M. Other convolution-based detection frameworks are only trained on the training set of FAIR1M. Meanwhile, this paper uses the AdamW optimizer to train DFCformer. The initial learning rate is set to 0.0001, and the weight attenuation is 0.005. First, 6 epochs are taken for warm-up training, and the learning rate is updated by one-dimensional linear interpolation. After warm-up training, the cosine annealing function updates the learning rate. The parameters of each part of the loss function are dynamically updated in the training process. Finally, the DFCformer becomes stable at 71 epochs with a batch size of 32, and the focusing parameters $v$ and $\varsigma$ are 0.6 and 0.5, respectively.

*4.4. Comparison with the State-of-the-Art*

Our method is compared with several counterparts, including CorrNet [20], ClusDet [16], GLSAN [19], M-CenterNet [17], DRN [44], TPH-YOLOv5 [28], and O$^2$DETR [48]. Table 2 summarizes the detection sub-classification results. Overall, DFCformer has significant performance advantages over other networks, with a mAP of 40.14% and an mAP$_{\text{F}}$ of 13.12%, respectively. Although other networks obtain similar results on mAP, the results of mAP$_{\text{F}}$ show the difference in detection ability for similar objects.

In addition, the results of each subcategory have obvious discrimination, which meets the distribution of instances in each subcategory in Figure 12. Specifically, for objects with a large scale and apparent features, such as small cars and intersections, the networks perform well and can make accurate detections. On the contrary, the networks show poor performance and low feature classification scores for instances with a small volume, such as ARJ21 and C919. Further, due to the transformer's intense need for data, the ability of the DFCformer to detect few-shot objects is lowest. Scarce objects and fine-grained objects are always challenging for visual recognition. In this case, the volume of data limits the evolution of network capacity, and the network is in a severe overfitting state. Meanwhile, the network's sensitivity to subtle key features is too low, so it is difficult to capture the features of the objects and accurately identify the category of the objects.

**Table 2.** The results on the FAIR1M dataset.

| General Categories | Categories | CorrNet [20] AP/AP$_F$ (%) | ClusDet [16] AP/AP$_F$ (%) | GLSAN [19] AP/AP$_F$ (%) | M-CenterNet [17] AP/AP$_F$ (%) | TPH-YOLOv5 [28] AP/AP$_F$ (%) | O$^2$DETR [48] AP/AP$_F$ (%) | DFCformer AP/AP$_F$ (%) |
|---|---|---|---|---|---|---|---|---|
| Road | Bridge | 15.71/7.65 | 14.65/8.21 | 13.18/7.66 | 15.16/8.33 | 14.83/7.12 | 17.61/7.95 | 18.67/8.02 |
| | Roundabout | 21.15/6.89 | 20.17/6.53 | 22.33/7.97 | 23.15/8.09 | 20.15/8.49 | 19.51/8.05 | 24.51/8.37 |
| | Intersection | 63.86/11.2 | 67.45/15.64 | 69.88/10.15 | 70.13/16.68 | 66.89/14.86 | 65.31/14.63 | 71.64/21.02 |
| Court | Baseball Field | 69.5/11.5 | 66.8/13.46 | 65.27/15.11 | 65.33/14.49 | 70.15/19.68 | 69.94/19.34 | 70.05/22.67 |
| | Tennis Court | 85.6/19.6 | 82.97/21.11 | 83.39/23.16 | 81.16/22.98 | 88.64/27.61 | 88.51/27.53 | 89.15/28.15 |
| | Football Field | 57.64/11.7 | 55.86/10.68 | 57.96/12.87 | 58.84/13.39 | 61.12/17.38 | 62.28/17.44 | 64.15/18.12 |
| | Basketball Field | 55.62/11.2 | 59.87/13.63 | 60.12/18.65 | 61.11/20.09 | 63.48/19.84 | 64.51/19.93 | 65.77/20.15 |
| Vehicle | Bus | 25.16/10.8 | 23.12/9.06 | 19.14/5.23 | 20.07/9.58 | 23.16/8.73 | 22.02/8.62 | 38.61/15.28 |
| | Van | 58.62/15.81 | 59.64/18.64 | 61.16/19.62 | 58.87/17.74 | 62.43/17.62 | 61.15/16.54 | 67.84/19.57 |
| | Trailer | 23.34/10.68 | 22.99/11.63 | 23.51/11.77 | 21.08/9.84 | 23.53/6.88 | 23.49/6.53 | 26.34/7.88 |
| | Tractor | 9.98/2.68 | 10.08/3.53 | 11.29/4.64 | 12.21/4.06 | 15.83/4.03 | 16.77/4.58 | 7.15/1.17 |
| | Excavator | 21.08/10.65 | 23.96/11.94 | 22.03/10.98 | 21.88/9.67 | 22.15/3.19 | 22.03/3.02 | 27.98/8.12 |
| | Small Car | 71.35/18.15 | 73.81/19.08 | 71.19/20.03 | 70.05/19.63 | 76.62/20.05 | 76.13/19.95 | 78.65/25.33 |
| | Cargo Truck | 50.32/12.42 | 51.21/15.89 | 52.21/17.83 | 50.56/18.34 | 52.27/18.62 | 50.06/18.12 | 55.87/23.17 |
| | Dump Truck | 41.63/10.35 | 45.19/12.37 | 47.55/13.38 | 43.15/11.89 | 49.31/16.15 | 47.35/15.48 | 49.86/16.53 |
| | Truck Tractor | 33.68/12.92 | 35.86/11.14 | 34.19/10.25 | 30.06/9.59 | 37.14/12.28 | 36.41/11.73 | 39.97/13.08 |
| | Other-vehicle | 13.86/5.91 | 15.12/7.26 | 17.62/8.21 | 15.31/8.87 | 15.09/6.34 | 15.98/6.87 | 15.12/7.15 |
| Ship | Warship | 23.41/10.38 | 20.17/9.64 | 19.56/9.25 | 20.06/8.93 | 23.98/7.57 | 22.08/7.21 | 27.88/9.54 |
| | Tugboat | 24.89/10.56 | 22.35/9.12 | 20.11/8.64 | 19.59/7.32 | 24.64/7.63 | 26.18/8.05 | 30.54/9.35 |
| | Motorboat | 28.62/12.62 | 25.64/11.05 | 24.66/9.25 | 23.65/8.74 | 30.04/9.83 | 31.25/9.95 | 39.88/10.05 |
| | Passenger Ship | 13.62/6.17 | 9.52/2.51 | 10.67/3.84 | 9.93/2.08 | 15.09/6.35 | 16.17/6.68 | 19.97/7.41 |
| | Fishing Boat | 5.21/1.84 | 3.12/0.87 | 5.11/1.12 | 6.09/1.46 | 3.21/1.01 | 3.05/0.87 | 1.08/0.15 |
| | Engineering Ship | 23.31/10.96 | 25.16/11.26 | 22.86/9.55 | 13.65/2.18 | 12.05/2.97 | 10.25/2.18 | 28.97/12.15 |
| | Liquid Cargo Ship | 17.86/12.61 | 15.64/8.76 | 13.06/7.74 | 5.34/0.63 | 12.75/3.16 | 11.34/2.75 | 15.35/3.45 |
| | Dry Cargo Ship | 29.86/12.37 | 30.16/13.62 | 28.62/12.19 | 22.61/7.15 | 25.09/5.93 | 22.97/5.36 | 24.15/4.51 |
| | other-ship | 6.11/2.86 | 5.11/1.29 | 3.26/0.25 | 4.88/1.01 | 4.35/0.61 | 4.18/0.86 | 2.86/0.48 |

**Table 2.** *Cont.*

| General Categories | Categories | CorrNet [20] | ClusDet [16] | GLSAN [19] | M-CenterNet [17] | TPH-YOLOv5 [28] | O$^2$DETR [48] | DFCformer |
|---|---|---|---|---|---|---|---|---|
| | | AP/AP$_F$ (%) | AP/AP$_F$ (%) | AP/AP$_F$ (%) | AP/AP$_F$ (%) | AP/AP$_F$ (%) | AP/AP$_F$ (%) | AP/AP$_F$ (%) |
| | C919 | 2.56/1.86 | 5.87/1.75 | 6.51/1.61 | 3.36/0.41 | 2.18/0.37 | 2.01/0.53 | 0.95/0.08 |
| | A220 | 45.62/16.55 | 44.12/18.31 | 47.12/19.58 | 49.62/15.32 | 51.86/19.96 | 50.54/19.17 | 55.37/20.09 |
| | A321 | 43.98/12.35 | 45.86/14.12 | 46.16/15.02 | 41.15/13.68 | 42.15/15.52 | 43.66/15.95 | 57.86/21.15 |
| | A330 | 22.14/6.85 | 27.85/7.86 | 26.68/8.05 | 21.67/8.84 | 26.04/6.53 | 25.37/6.28 | 31.17/7.68 |
| | A350 | 22.69/8.31 | 22.83/9.37 | 21.16/7.78 | 23.36/7.42 | 25.12/5.01 | 26.12/5.43 | 35.57/8.69 |
| Airplane | ARJ21 | 3.69/0.98 | 5.89/0.46 | 6.09/1.05 | 7.05/1.88 | 9.54/2.26 | 9.36/2.09 | 2.84/0.61 |
| | Boeing737 | 42.67/15.58 | 45.98/17.67 | 44.89/18.52 | 46.94/15.61 | 48.68/16.83 | 49.15/16.97 | 48.97/17.02 |
| | Boeing747 | 58.98/23.19 | 61.31/22.18 | 63.33/23.14 | 67.95/24.57 | 69.91/24.16 | 70.18/24.83 | 75.56/27.86 |
| | Boeing777 | 27.98/7.18 | 25.46/9.56 | 28.96/6.88 | 29.05/6.34 | 27.58/7.53 | 28.69/7.74 | 31.25/11.37 |
| | Boeing787 | 55.64/16.67 | 53.88/19.95 | 57.97/20.18 | 59.91/18.86 | 61.86/19.94 | 63.87/20.26 | 69.87/22.16 |
| | other-airplane | 65.32/20.16 | 68.52/20.47 | 69.88/21.83 | 67.35/23.81 | 67.82/24.35 | 68.94/25.23 | 73.69/27.68 |
| *mAP/mAP$_F$* | | 34.66/10.82 | 34.95/11.34 | 35.1/11.24 | 34.09/10.81 | 36.41/11.25 | 36.34/10.94 | 40.14/13.12 |

Table 2 shows the test comparison results of the network on the FAIR1M dataset. On this basis, we compare the accuracy and speed of different networks in Figure 14. Regarding reasoning speed, CorrNet and TPH-YOLOv5 have significant advantages, reaching 31 FPS and 29 FPS, respectively. However, aerial target detection is an engineering application task. The balance between network reasoning speed and performance determines the ability of the network to solve the actual aerial target detection task. Therefore, we should make a comprehensive evaluation from two aspects: reasoning speed and detection accuracy. Although the reasoning speed of the DFCformer does not exceed that of CoorNet and TPH-YOLOv5, its 27 FPS reasoning speed still has certain advantages over the rest of the networks. More importantly, its detection accuracy has apparent advantages over other networks. By comprehensively analyzing the performance of reasoning speed and detection accuracy, the DFCformer has a solid ability to solve aerial target detection tasks.

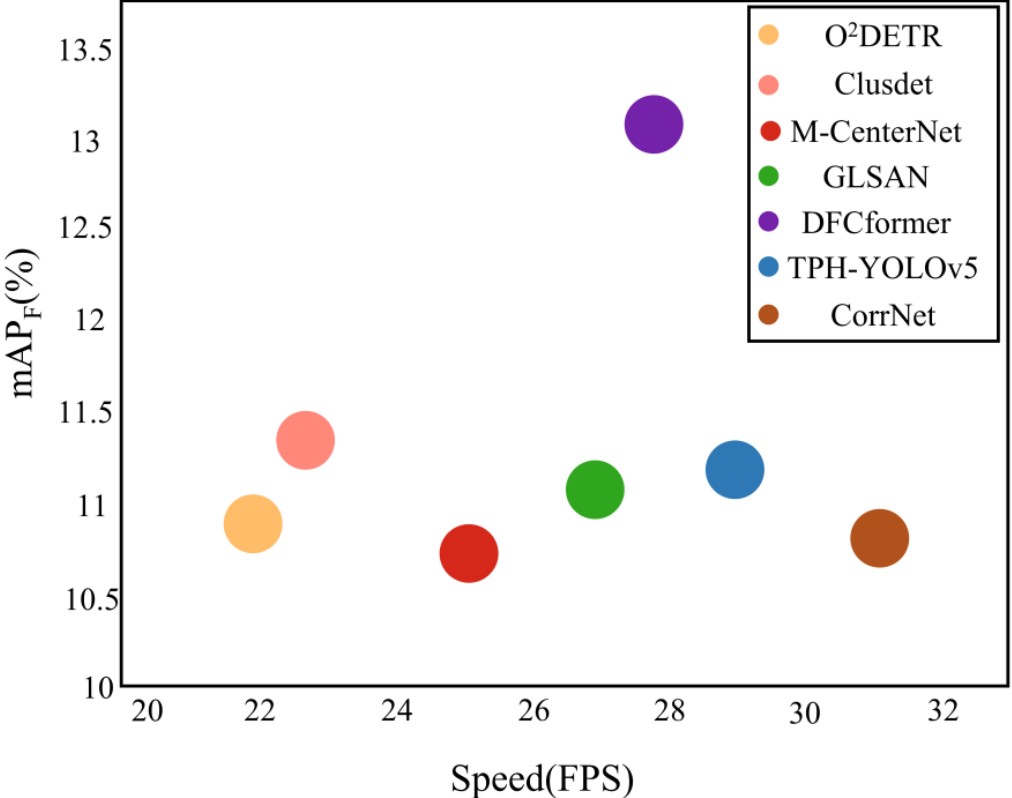

**Figure 14.** Performance versus speed on the FAIR1M data set.

Figure 15 shows the detection results of the networks in three aerial scenes. Figure 15a shows the aerial images of urban residential areas with complex backgrounds. In this scene, the automobile, truck, and other examples are distributed along the road. Due to their tiny pixel proportion, most networks in the experiment fail to capture compelling features to make accurate detections. The proposed DFCformer can only alleviate the limitation of minimal object feature extraction to a certain extent, and it is difficult to identify different types of vehicles accurately. The yellow dotted box in the figure indicates the complex area in the scene. The baseball field will be attached to the surrounding training area in the actual construction, which increases the inter-domain span between the natural setting and the training samples, and the network's generalization ability is insufficient to overcome such an inter-domain span. All networks except DFCformer mistakenly detect the training field of the baseball field as a football field. DFCformer can overcome the inter-domain span migration, but it still fails to reflect the accurate edge distribution of the baseball field.

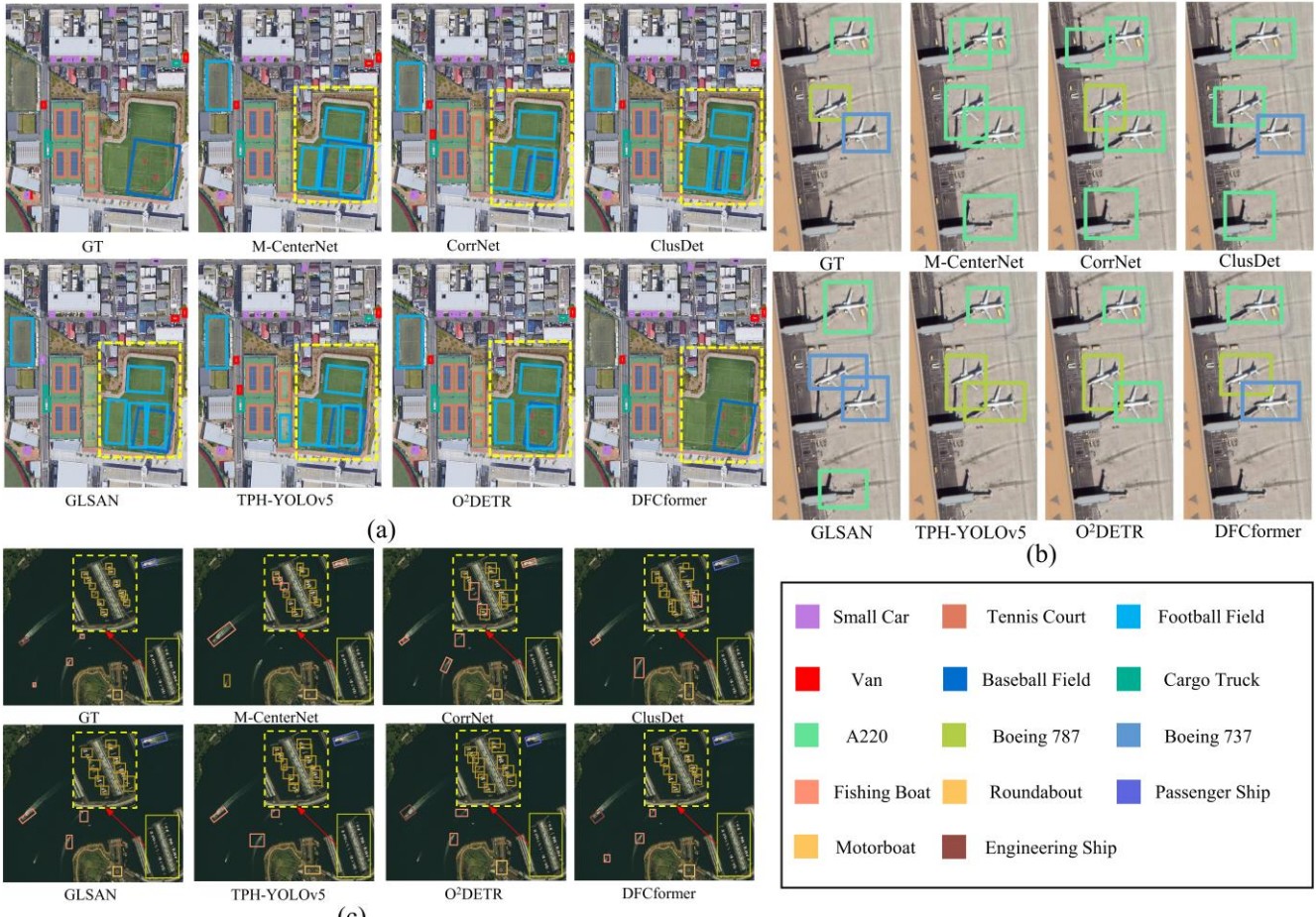

**Figure 15.** Results of the networks in three aerial scenes. (**a**) aerial image of urban residential areas with complex backgrounds. (**b**) aerial image of the airport. (**c**) aerial image of the water area.

Figure 15b is an aerial image of the airport. This scene has a simple background, and the objects are distributed longitudinally along with the terminal. All networks can detect airplanes, but different networks have significant differences in the subdivision ability of similar objects. The convolution-based networks (M-CenterNet, CorrNet, ClusDet, and GLSAN) focus on acquiring local features but ignore the establishment of long-distance feature mapping. So, these networks cannot classify different objects accurately and even mistakenly detect the terminal as A220. On the contrary, the transformer-based networks focus on global feature information processing to capture rich features, and they can accurately identify similar objects.

Figure 15c shows an aerial image of the water area. It is challenging for the network to realize the fine classification of fishing boats, cruise ships, and motorboats. The similarity between the classes is still a challenge for the aerial detectors. Besides, the yellow dotted line area in the scene shows densely distributed objects with a small proportion of pixels, and almost all networks cannot accurately detect the objects in this area. In contrast, the proposed DFCformer is superior to other networks in terms of locating accuracy, classification accuracy, and prediction box quality.

The above analysis and the test results in Table 2 indicate that DFCformer has higher regression accuracy and classification scores than other networks, indicating that the transformer-based backbone network, feature reconciliation guidance component, and cascaded attentional interactive module in this paper can alleviate the limitations of current detection to a certain extent, In the follow-up work, we will further improve the fine classification ability of the network and improve the accuracy of similar object detection.

*4.5. Ablation Study*

The comparative experiment in Section 4.4 shows the excellent detection performance of the DFCformer. To further verify the effectiveness of DFCformer and the contribution of each component to the overall network, this section sets up ablation experiments under the same experimental conditions.

**Analysis of the effectiveness of DMViT.** To evaluate the effectiveness and the feature interpretation mechanism of DMViT, this paper chooses two standard backbones (ResNet101 [68] and PVT [29]) as substitutes for DMViT. As shown in Table 3, the experimental results of different combinations are different. Specifically, the mAP of taking DMViT as the backbone is 3.23% and 5.13% higher than that of taking PVT and ResNet101 as the backbone. Meanwhile, the $mAP_F$ of taking DMViT as the backbone is the best, which is about 3.1% higher than the worst.

**Table 3.** The results of the ablation study on different component combinations.

| Combination | $mAP$ | $mAP_F$ |
|---|---|---|
| ResNet101 + FRGC + CAIM | 34.83 | 10.06 |
| PVT + FRGC + CAIM | 36.85 | 12.57 |
| DMViT + FRGC + CAIM (DFCformer) | 40.08 | 13.17 |

Figure 16 shows the experimental results in different scenes. By comprehensively analyzing the results in the three scenes, due to the limited receptive field, ResNet101 only captures the local feature information. On the contrary, PVT only focuses on the global feature mapping and ignores the detailed information, thus reducing the separability of background and foreground. Due to the influence of deformable patches and multi-scale attention, DMViT considers multi-scale long-distance modeling and focuses on local detail information, so it can obtain fine-grained information of multi-scale instances at the global level and capture rich instance features.

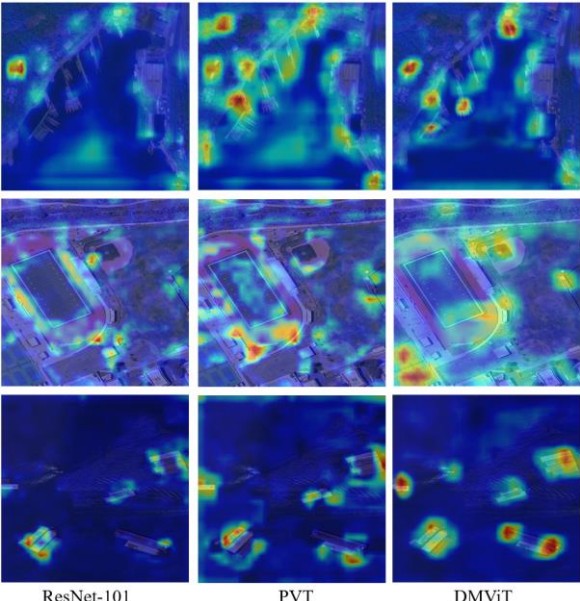

**Figure 16.** Comparisons with different backbone networks.

A strong backbone plays an irreplaceable role in aerial object detection networks. DMViT makes an outstanding contribution to the overall performance of DFCformer with its comprehensive and accurate feature expression ability.

**Analysis of the effectiveness of FRGC.** To verify the effectiveness and the mechanism of FRGC, this paper replaces FRGC with the ordinary linear interpolation sampling

operation (S) to process the features of adjacent layers. Considering the experimental cycle and the characteristics of FRGC's feature processing, this paper separately takes the airplane sub-dataset in FAIR1M as the data source of this experiment. Then, two groups of experimental results are counted and presented in the form of a confusion matrix. As shown in Figure 17, after replacing FRGC with a linear interpolation sampling operation, the detection accuracy of specific examples decreases seriously, and the largest accuracy decrease reaches 23%. These degradations show that the simple sampling operation cannot efficiently deal with different feature layers. Also, the insufficient attention to fine-grained details and the imbalance of the sample number limit the network detection performance.

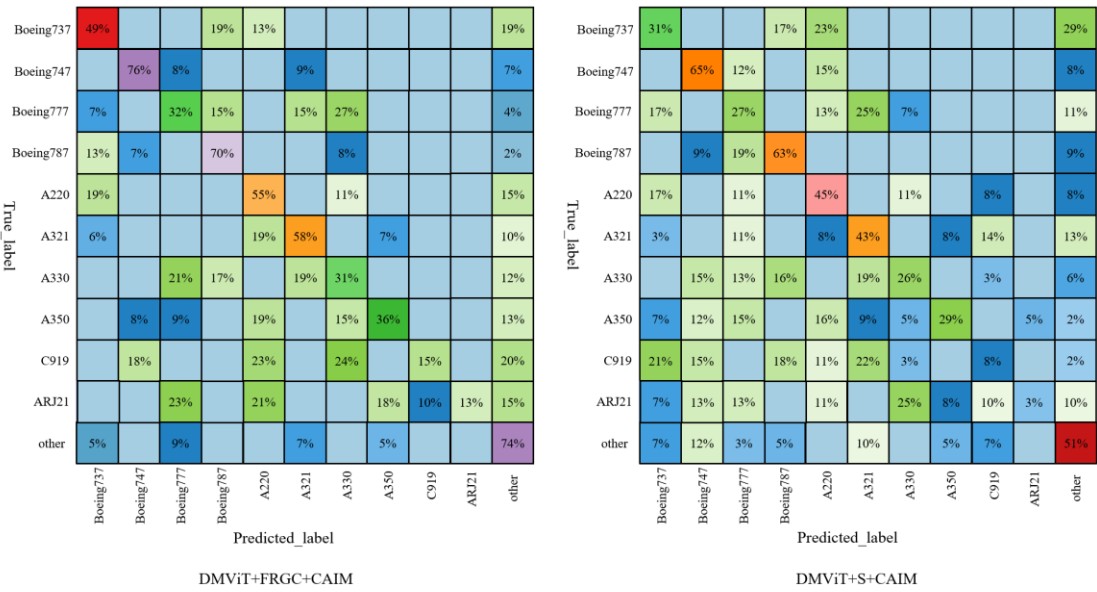

**Figure 17.** The confusion matrices of the variants on the airplane sub-dataset.

**Analysis of the effectiveness of CAIM.** To study the impact of the CAIM, this paper replaces the CAIM with standard feature fusion approaches (concatenation and addition) in the DFCformer. Figure 18 shows the comparison results. Comparing concatenation and addition, it can be found that our CAIM performs better in the two sub-datasets. The superior performances indicate that the CAIM helps to improve the multi-scale aerial object detection ability of the network.

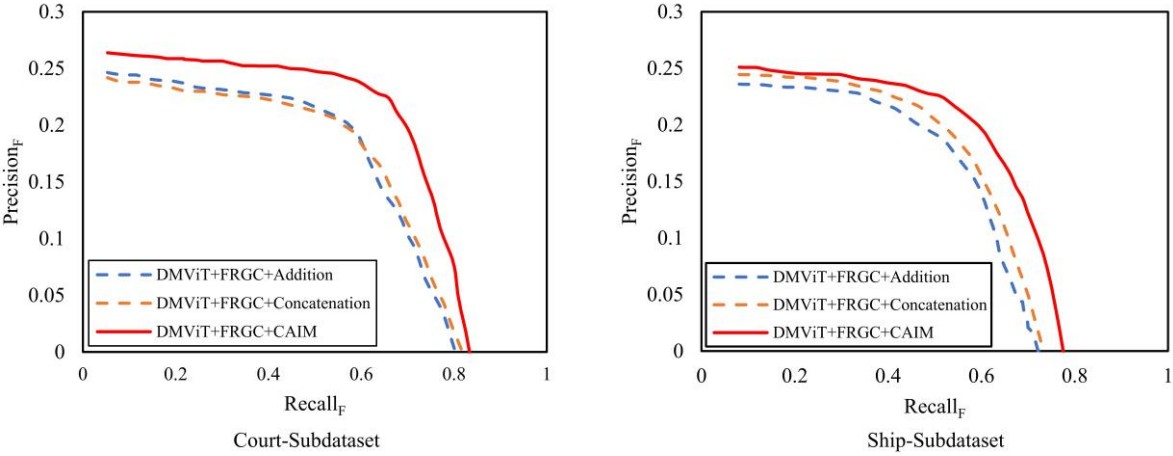

**Figure 18.** Comparison results with $PR_F$-curve on the two sub-datasets.

## 5. Conclusions

To overcome the challenges in aerial object detection such as large-scale variations and confused distinguishing features, this paper proposes a novel and powerful aerial object detection framework called DFCformer. The whole framework includes three parts: DMViT, FRGC, and CAIM. DMViT is the backbone, which introduces deformation patch embedding and multi-scale adaptive self-attention to improve the ability to capture small-scale objects in aerial images with a large field of view. FRGC is the feature coordination and guidance component, which guides feature interaction layer by layer according to the focuses of feature semantic information expression at different levels. CAIM is the cascade attention interactive module, which adopts a cascade attention fusion strategy to perform hierarchical reasoning on the relationship between different levels of features. Finally, a serious of experiments are conducted on the FAIR1M, and the DFCformer achieves the best performance, highlighting its effectiveness. The disadvantage of DFCformer is that due to the use of a transformer-based network, the minimum size of configuration parameters reaches 65M, which is not easy for unmanned aerial vehicle platforms. Our future works will lightweight the network. We hope this attempt could promote the development of fine-grained object recognition in aerial scenes and broaden the application scope of the transformer.

**Author Contributions:** Conceptualization, X.H. and Q.W.; methodology, F.S.; software, H.W.; validation, G.L. and J.W.; formal analysis, G.L.; investigation, G.L.; resources, J.W.; data curation, G.L.; writing—original draft preparation, G.L.; writing—review and editing, G.L.; visualization, X.H.; supervision, F.S.; project administration, Q.W.; funding acquisition, F.S. All authors have read and agreed to the published version of the manuscript.

**Funding:** This research was funded by the National Natural Science Foundation of China (No.61671470), and the Key Research and Development Program of China (No. 2016YFC0802900).

**Institutional Review Board Statement:** Not applicable.

**Informed Consent Statement:** Not applicable.

**Data Availability Statement:** The data in the study comes from the public data set, and its link address is gaofen-challenge.com [11 May 2022].

**Conflicts of Interest:** The authors declare no conflict of interest.

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
