# Peer review of "A Novel Multi-Scale Transformer for Object Detection in Aerial Scenes"

_drones, doi:10.3390/drones6080188_

Round 1

Reviewer 1 Report

This paper proposes a nove aerial object detection framework called DFCformer to address challenges of the large-scale variation of objects and the confusion of category features. Extensive experiments have been conducted on the dataset, and DFCformer shows its advantages by achieving the highest scores with stronger scene adaptability. The purpose of this paper is to solve the practical problems in remote sensing target detection, the organization is rigorous and the formula derivation is reasonable. However, there are still some problem (already mentioned in the comments). 

1. line 35-38: The difference of scale in the perspective of remote sensing is mainly due to inherent scale differences between different instances in the same scene.

2. In the existing work, whether there is any work to introduce attention mechanism into Transformer, if so, please review.

3. In the related work, you should emphasize how the proposed methods of this paper is constructed and the relationship with these related works.

4. In figure 3, the abbreviation of "deformable patch embedding" should be "Depatch Emb.".

5. CAIM should be briefly introduced in the title of Figure 3.

6. The parameters Woffset and Wscale in Equation 5 and 6 are initialized to 0, how do they backpropagate, and should they be initialized to 1?

7. Some feature map F from line 392 to 395 need to be marked in the Figure 9, This is more convenient for readers to understand.

8. In figure 10, please indicate which line is FLOPS and which is Top-1.

9. The method proposed in this paper is an improvement for large-scale differences. In line558-560, what is the problem with the poor recognition performance of small volume ARJ21 and C919? whether to prove that the multi-scale performance of the model needs to be improved?

10. Some figure captions, such as Figure 8 and 9 are shallow and they must be improved.

11. In line 18 to 19, ‘Extensive experiments have been conducted on 18 the dataset, …’, It is best to specify the full name of the dataset.

12. In section 4.2, you use FIoU as Evaluations Metrics. Whether FIoU is used for testing only, or for monitoring bbox regression. And you should analyze the role of FIoU in ablation experiments.

Reviewer 2 Report

The authors proposed a Transformer-based framework to resolve multiscale-caused issues. The three components methods showed the capability to deal with various scales and circumstances.

This manuscript is easy to read and mostly friendly to follow and comprehend the authors’ targets and focus. However, I got some questions/advice for the authors:

- I didn’t recognize the inference time in this manuscript. Since infer-time is essential to UAV object detection task, could you illustrate it as well?

- Why don’t you compare with other SOTAs, such as TPH-YOLOv5. You might have noticed that this improved yolov5 is also based on the Transformer to settle multi-scale problems. Could you add several experiments on this?

- Have you considered employing the scheme at the drone-end?

Round 2

Reviewer 1 Report

The authors have addressed the reviewer comments.